

# Proteomic detection of COX-2 pathway-related factors in patients with adenomyosis

Jihua Zhang*, Luying Shi*, Jingya Duan, Minmin Li and Canyu Li

Department of Gynecology, the Third Affiliated Hospital of Zhengzhou University, Zhengzhou, Henan, China

* These authors contributed equally to this work.

## ABSTRACT

**Background:** Investigating the relationship between cyclooxygenase-2 (COX-2) pathway-related factors and clinical features in patients with adenomyosis by proteomics could provide potential therapeutic targets.

**Methods:** This study recruited 40 patients undergoing surgical hysterectomy and pathological diagnosis of adenomyosis, collected ectopic endometrial specimens, and recorded clinical data. The expression levels of COX-2 in ectopic uterus lesions were detected using the immunohistochemical (IHC) SP method. The 40 samples were then divided into a COX-2 low or high expression group. Five samples with the most typical expression levels were selected from each of the two groups and the differential proteins between the two groups were identified using label-free quantitative proteomics. WW domain-binding protein 2 (WBP2), interferon induced transmembrane protein 3 (IFITM3), and secreted frizzled-related protein 4 (SFRP4) were selected for further verification, and their relationships with COX-2 and clinical characteristics were analyzed.

**Results:** There were statistically significant differences in the expression of WBP2, IFITM3, and SFRP4 between the COX-2 low and high expression groups ($P < 0.01$). The expressions of COX-2, IFITM3, and SFRP4 were significantly correlated with dysmenorrhea between the two groups ($P < 0.05$), but not with uterine size or menstrual volume ($P > 0.05$). However, there was no significant correlation between the expression of WBP2 and dysmenorrhea, uterine size, and menstruation volume in both the high expression and low expression groups ($P > 0.05$).

**Conclusions:** COX-2, IFITM3, SFRP4, and WBP2 may be involved in the pathogenesis of adenomyosis. COX-2, IFITM3, and SFRP4 may serve as potential molecular biomarkers or therapeutic targets in dysmenorrhea in patients with early adenomyosis.

## INTRODUCTION

Adenomyosis is a benign pathologic state characterized by the presence of ectopic endometrial glands and stroma in the diffuse or focal myometrium. The etiology and pathogenesis of adenomyosis remain unclear. Some theories have explained the

Corresponding author
Canyu Li, lcy200507@zzu.edu.cn

pathophysiological mechanisms of adenomyosis, with the most widespread hypothesis being that adenomyosis originates from endometrial glands and stromal cells invading the myometrium. The clinical symptoms of adenomyosis include progressive dysmenorrhea, an enlarged uterus, heavy menstrual bleeding, and infertility (*García-Solares et al., 2018*; *Zhai et al., 2020*). Understanding the underlying mechanisms of key molecules associated with the development of adenomyosis may provide a new approach to the targeted treatment of the condition.

It has been suggested that inflammatory mediators, neuroangiogenesis, cell proliferation, and intimal and interstitial invasion may be associated with the occurrence, development, and dysmenorrhea symptoms of adenomyosis (*Vannuccini et al., 2017*; *An et al., 2017*). Nonsteroidal anti-inflammatory drugs (NSAIDs) are part of first line of treatment for adenomyosis (*Kho, Chen & Halvorson, 2021*). NSAIDs inhibit COX to prevent the synthesis of prostaglandin E2 (PGE2), and are widely available for treating primary and secondary dysmenorrhea symptoms (*Marjoribanks et al., 2015*). COX-2 is a key factor in the inflammatory pathway (*AlAshqar et al., 2021*) and COX-2 expression levels correlate with the degree of dysmenorrhoea in adenomyosis (*Li et al., 2019*). Previous studies show that COX-2 is overexpressed in mesenchymal stem cells originating from the endometrium of adenomyosis lesions and normal uterus, and a COX-2 inhibitor can inhibit the invasion and metastasis of mesenchymal stem cells derived from adenomyosis and induce apoptosis (*Chen et al., 2010*). Our previous studies have also found that silencing COX-2 expression can inhibit stromal cell proliferation and invasion, and inhibiting COX-2 can reduce the depth of endometrial infiltration and the level of vascular growth factor (VEGF) in the uterine tissue of mice with adenomyosis (*Liang et al., 2021*). Additionally, COX-2 promotes cell proliferation through the Wnt/β-catenin signaling pathway in a variety of cancers (*Buchanan & DuBois, 2006*; *Xu et al., 2018*). However, the mechanism by which COX-2 affects endometrial cells remains unclear.

The ongoing development of label-free proteomic quantitation has empowered the ability to characterize and quantify tissue proteomes for clinically relevant activities, such as identifying biomarkers (*Wiśniewski, Ostasiewicz & Mann, 2011*). The present study identified differentially expressed proteins between the COX-2 low expression and high expression groups using label-free quantitative proteomics. The distribution of related proteins in adenomyosis tissue was then validated with IHC and their relationships with clinical features were studied. This study aimed to discover target genes associated with adenomyosis that may serve as indicators for early prediction or condition monitoring, as well as potential targets for treatment.

## MATERIALS AND METHODS

### Tissue samples

A total of 40 patients who underwent hysterectomy and were pathologically diagnosed with adenomyosis at the Third Affiliated Hospital of Zhengzhou University were enrolled in the study, on the basis of the following criteria: (i) their uterine samples were verified by pathology; (ii) they were reproductive age or premenopausal women between the ages of 30 to 55 years who had no malignant tumors and chronic disease; and (iii) they did not

**Table 1 Clinical features of patients with adenomyosis.**

| Characteristics | Overall population | No. of cases $n$ (%) |
|---|---|---|
| Age ($\bar{x} \pm SD$), y | 47.15 ± 4.19 | |
| | 30–45 | 15 (37.5) |
| | 46–55 | 25 (62.5) |
| Menstrual cycle | | |
| | Proliferative period | 25 (62.5) |
| | Secretory period | 15 (37.5) |
| Dysmenorrhea (score) | | |
| | No/mild | 7 (17.5) |
| | Moderate | 24 (60.0) |
| | Severe | 9 (22.5) |
| Uterine volume ($\bar{x} \pm SD$), cm$^3$ | 311.93 ± 192.88 | |
| | Normal | 27 (67.5) |
| | Increscent | 13 (32.5) |
| Menstrual volume (ml) | | |
| | Normal | 12 (30.0) |
| | Increscent | 28 (70.0) |

Notes:
n, number
$\bar{x}$: mean
SD: standard deviation.

receive any hormone therapy in the 6 months prior to surgery. After sample collection, part of each specimen was stored at −80 °C for further processing. The rest of the samples were fixed in 4% formaldehyde solution, routinely dehydrated, and then embedded in paraffin for IHC staining. The protocol for this study was approved by the Ethics Committee of The Third Affiliated Hospital of Zhengzhou University. Written informed consent was obtained from all study participants in accordance with the Declaration of Helsinki.

## Clinical data collection

The clinical data of all patients were obtained by questionnaire before surgery. These data included: patient age, menstrual cycle, visual analogue score (VAS), menstrual blood volume, and volume (V) of the uterine (calculated by the formula: V [cm$^3$] = length × width × height × 0.523, with greater than 350 cm$^3$ regarded as uterine enlargement). The VAS scores were grouped into three levels: no dysmenorrhea (0), mild (1–3), moderate (4–6), and severe (7–10). Menstrual blood loss was estimated using a pictorial blood loss assessment chart (PBAC). A chart score >100 was equivalent to blood loss >80 ml (Table 1).

## Immunohistochemistry

Rabbit-derived antibodies specific to COX-2 (1:50), SFRP4 (1:400;), IFITM3 (1:600), and WBP2 (1:100) were purchased from ProteintechGroup, lnc., Wuhan, China. The IHC kit was purchased from Neobioscience Technology Company (Beijing, China). Tissue sections were dewaxed and rehydrated, then antigen retrieval was implemented by boiling slices in

citrate buffer (0.01 M). The endogenous peroxidase was blocked using 3% $H_2O_2$. The samples were then treated with normal goat serum for 30 min at room temperature, incubated with primary antibodies at 37 °C for 30 min, and then treated with biotinylated secondary antibodies for 30 min at room temperature. Next, the sections were labeled with horseradish peroxidase for 30 min at room temperature. Sections were visualized after being incubated with diaminobenzidine (DAB). Last, immunostaining intensity (0, unstained; 1, light yellow; 2p, ale brown; and 3, tan) and the percentage of cells stained (0, <5%; 1, 5–25%; 2, 26–50%; 3, 51–75%; 4, 76–100%) were both evaluated. The two scores were then added to assess staining, with total scores as follows: 0, negative (-); 1 to 4, weakly positive (+); 5 to 8, positive (++); and 9 to 12, strongly positive (+ + +). The negative and the weakly positive were low expressions, and the positive and the strongly positive were high expressions. The stained slides were evaluated by two pathologists.

## Sample preparation and tryptic digestion

A portion of the sample was placed in a grinding and shaking tube and a suitable volume of lysis buffer (8 M urea, 1% SDS) was added. After being shaken three times for 40 s each time, the samples were subsequently lysed on ice for 30 min. The specimens were then centrifuged at $16,000 \times g$ at 4 °C for 10 min, and the retained supernatants were analyzed. Protein concentrations were quantified using a BCA protein assay kit (Thermo Fisher Scientific, Waltham, MA, USA). A total of 100 μg of protein samples, added to 100 mM triethylammonium bicarbonate buffer (TEAB; Sigma, Buchs, Switzerland), was reacted with 100 mM TCEP for 60 min at 37 °C. Alkylation with 40 mM iodoacetamide was performed for 40 min in the dark at room temperature. Protein samples were precipitated with cold acetone with an acetone-to-samples ratio of 6:1 for 4 h at −20 °C. Samples were centrifuged at $10,000 \times g$ for 20 min. The retained precipitate was dissolved with 100 μL 100 mM TEAB and then digested with trypsin at an enzyme-to-substrate rate of 1:50 for 12 h at 37 °C. Peptides were determined by a peptide quantification kit (Thermo Fisher Scientific, Waltham, MA, USA).

## LC–MS/MS analysis

Digested peptides were subjected to LC-MS/MS. Peptides were dissolved in buffer A (2% acetonitrile and 0.1% FA) and analyzed at a constant flow rate of 300 nL/min using an EASY-nLC 1200 (Thermo Fisher Scientific, Waltham, MA, USA) liquid chromatography system. Peptides were then separated on a reversed-phase C18 column (75 μm × 25 cm; Thermo Fisher Scientific, Waltham, MA, USA). The elution gradient was 5–23% of buffer B (0.1% FA in 98% ACN) for 103 min, 23–29% of buffer B for 11 min, 29–38% of buffer B for 2 min, 38–48% of buffer B for 2 min, 48–100% of buffer B for 2 min, and 100% of buffer B for 30 min. Separated peptides were analyzed by DDA (data dependent acquisition) mass spectrometry using a Q_Exactive (Thermo Fisher Scientific, Waltham, MA, USA) mass spectrometer. The MS data were analyzed using MaxQuant software version 1.6.2.10.

## Bioinformatics analysis

The proteins searched in the database (Uniprot) were analyzed, and then differentially expressed proteins were screened. Differentially expressed proteins were defined as meeting the following criterion: $P < 0.05$ and FC < 0.50 or FC > 2.0. The $P$-value of the significant difference between the samples was calculated using the t-test in the R language. A functional enrichment analysis was performed with the Database for Annotation, Visualization and Integrated Discovery (DAVID; https://david.ncifcrf.gov/) to determine the roles of differentially expressed proteins. A Gene Ontology (GO) analysis was performed to obtain significantly enriched terms to deduce important biological functions involving multiple proteins. A Kyoto Encyclopedia of Genes and Genomes (KEGG) pathway analysis was used to explore the pathways in which the differential proteins might be involved. Protein-protein interactions (PPIs) were analyzed with STRING (Search Tool for the Retrieval of Interacting Genes).

## Statistical analyses

The SPSS software version 26.0 (IBM Corp.) was used to perform statistical analyses. The enrichment analysis was determined by a two-tailed Fisher's exact test. Wilcoxon was used to compare factor expressions in the 40 samples divided into the high and low COX-2 expression groups. The correlations among the factors in the 40 samples were calculated using a Spearman rank correlation analysis. The relationship between the expression of factors and clinical characteristics in adenomyosis was analyzed by a chi-square test. All statistical graphs were drawn with GraphPad Prism 8.0. $P < 0.05$ indicated statistical significance.

# RESULTS

## IHC detection of COX-2

IHC staining suggested that COX-2 was mainly expressed in the cytoplasm and cell membrane of ectopic endometrial glandular cells in adenomyosis, without significant expression in the myometrium. The negative and the weakly positive were defined as the low expression group, and the positive and the strongly positive were the high expression group (Fig. 1A). Of the 40 tissues, nine cases were in the high expression group and 31 cases were in the low expression group. Five samples with the most typical expression were respectively selected from the high and low expression groups for further proteomics studies.

## Proteins identified in endometriosis tissue samples

A total of 278 differential proteins (93 upregulated and 185 downregulated proteins) were identified to be differentially expressed with a fold change >2.0 or <0.5, with 30 proteins expressed only in the high expression group and 56 proteins expressed only in the low expression group (Table 2). WBP2, IFITM3 (expressed only in the high expression group; Fig. 2A), and SFRP4 (expressed significantly differently between the two groups; Fig. 2B) were selected for verification. High expression of WBP2 is involved in the Wnt/β-catenin pathway and linked to cell proliferation and invasion in cancer tissues (*Tabatabaeian et al.,*
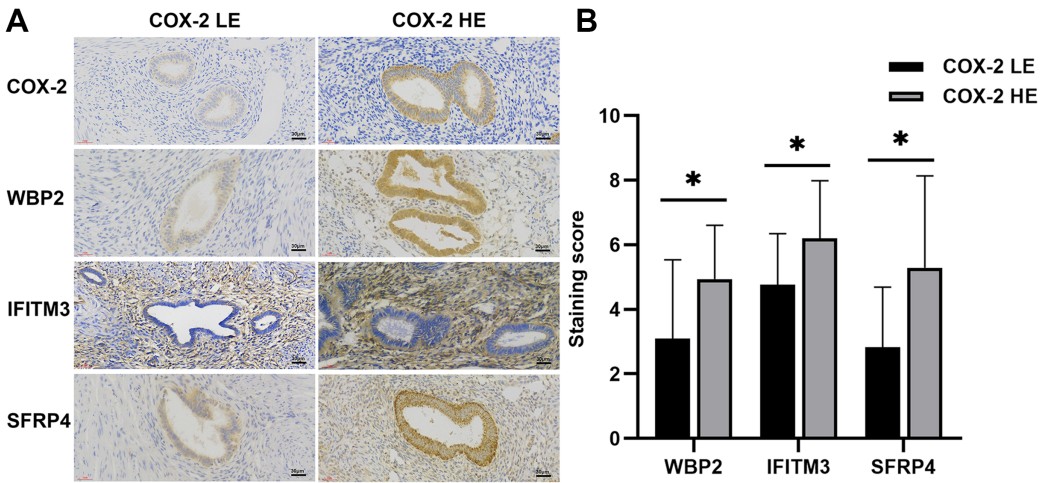

**Figure 1 IHC of COX-2 pathway-related factors in adenomyosis.** (A) Immunoexpression of COX-2, WBP2, IFITM3, and SFRP4. (B) Comparative immunoexpression of WBP2, IFITM3, and SFRP4 in COX-2 low and high expression groups. $^*P < 0.05$. Abbreviations: COX-2 HE, COX-2 high expression group; COX-2 LE, COX-2 low expression group.

**Table 2 Differential proteins were identified using label-free quantitative proteomics.**

| Differential proteins (A *vs* B) | All | Both | Differential | Up | Down | Only A | Only B | Neither |
|---|---|---|---|---|---|---|---|---|
| HE *vs* LE | 2,271 | 2,185 | 278 | 93 | 185 | 30 | 56 | 0 |

Notes:
HE: high expression group
LE: low expression group.

*2020*). IFITM3, which is significantly overexpressed in cancer tissues, promotes tumor invasion and migration, and is also engaged in epithelial-mesenchymal transition (EMT) by the Wnt pathway (*Rajapaksa, Jin & Dong, 2020*). SFRP4 has been documented to be associated with the Wnt pathway in a variety of cancers (*Deshmukh et al., 2019*; *Busuttil et al., 2021*). Ectopic endometrial cells in adenomyosis have abnormal proliferation, apoptosis, and invasion of the myometrium, and abnormal activation of the Wnt/β-catenin signaling pathway has been confirmed in adenomyosis tissue. A series of follow-up studies were then performed to better understand the relationships of the above factors with adenomyosis and with the Wnt/β-catenin pathway in adenomyosis.

## GO and KEGG pathway analyses and PPI network

GO analysis of differential proteins showed that the upregulated proteins were mainly enriched in localization of substances, intracellular components, and structural molecular activities (Fig. 2D), while the functional enrichment terms of downregulated proteins were mainly correlated with protein containing complex, multicellular organismal processes, and molecular transducer activity (Fig. 2E). From the KEGG pathway analysis, there were 10 pathways with the most meaningful enrichment of upregulated and downregulated proteins in the KEGG pathway. The pathways enriched in the highest number of upregulated proteins were influenza A, complement and coagulation cascade, apoptosis, staphylococcal aureus infection, and natural killer cell-mediated cytotoxic response

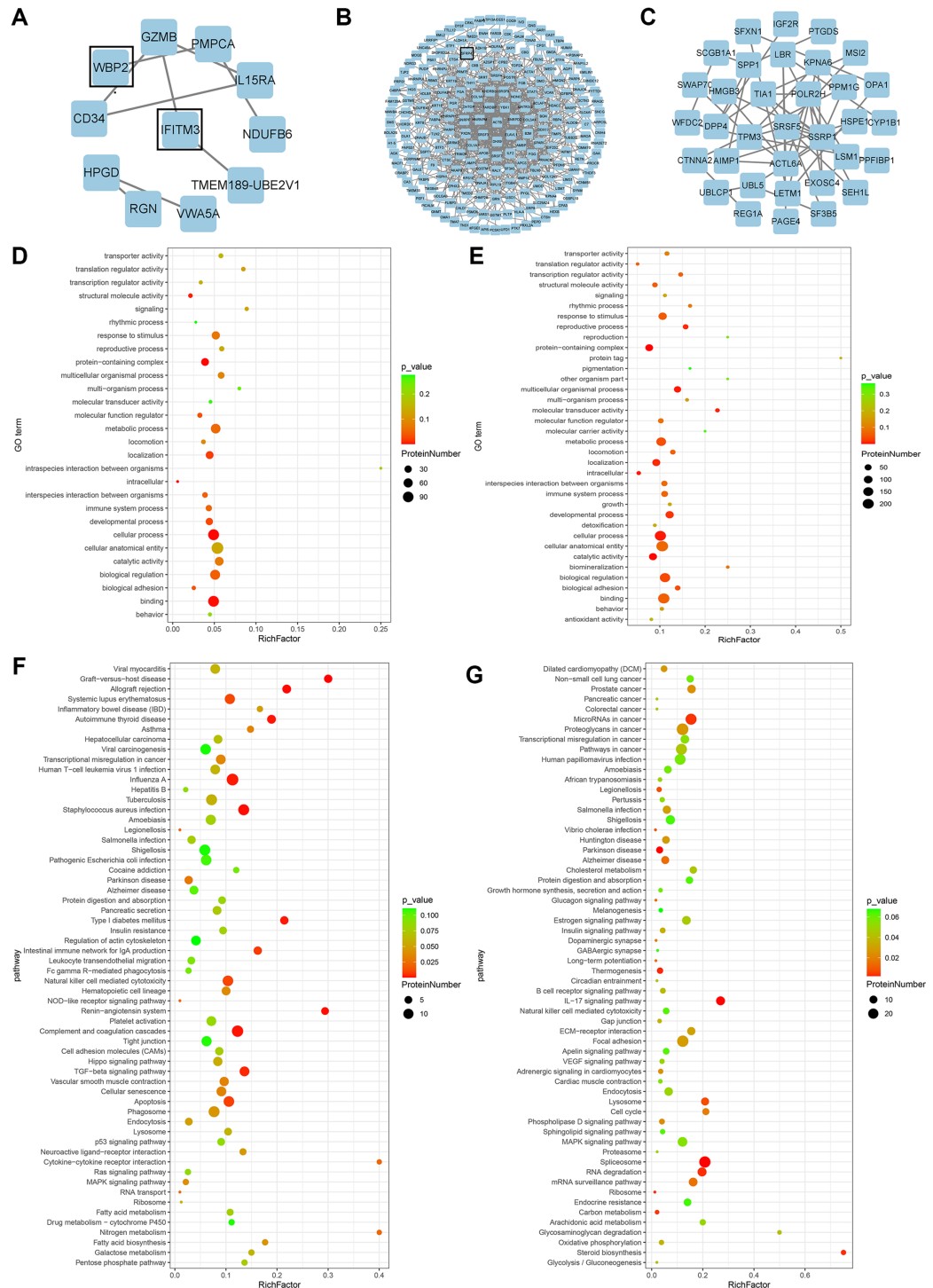

**Figure 2 Bioinformatics analysis.** (A) PPI network of proteins exclusively expressed in the high expression group. (B) PPI network of differential proteins expressed in both groups. (C) PPI network of proteins exclusively expressed in the low expression group. (D) GO enrichment analysis of up-regulated proteins. (E) GO enrichment analysis of down-regulated proteins. (F) KEGG enrichment analysis of up-regulated proteins. (G) KEGG enrichment analysis of down-regulated proteins.

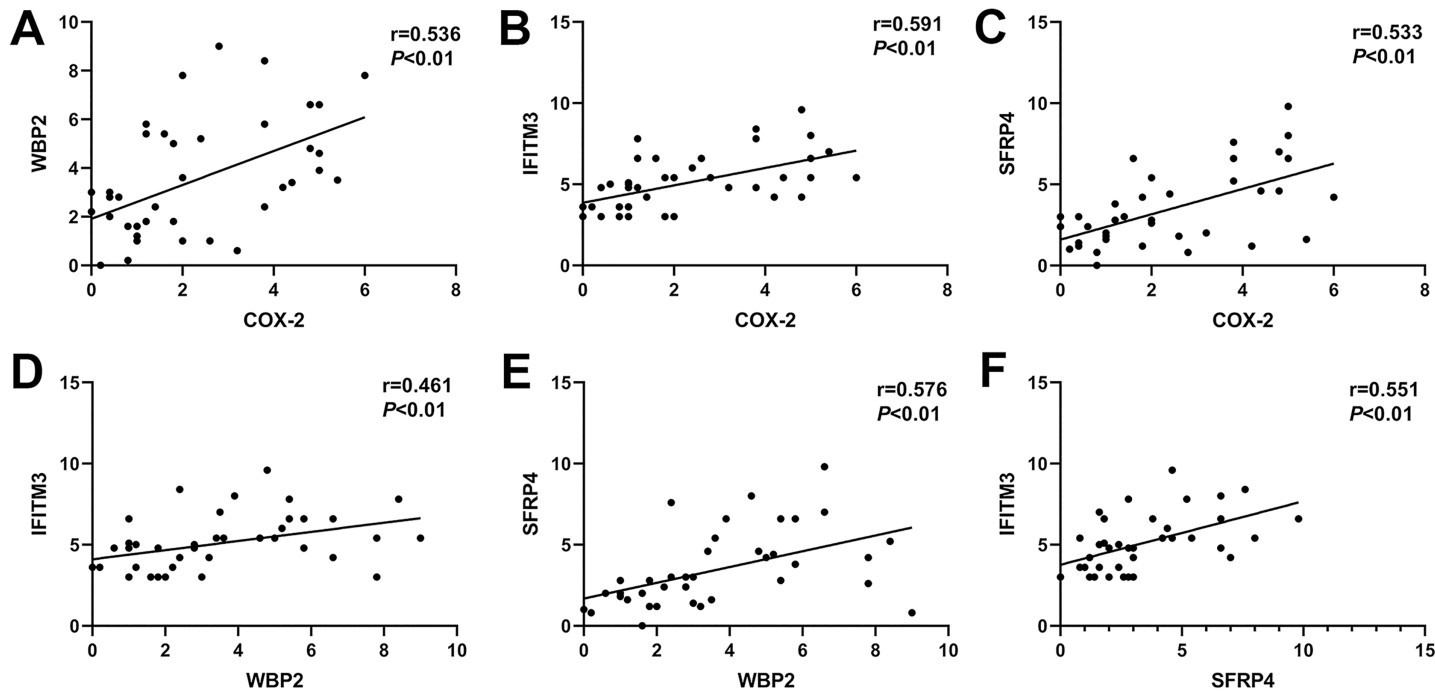

**Figure 3 Correlation of COX-2, SFRP4, WBP2, and IFITM3 expression in ectopic endometrium of adenomyosis.** (A) COX-2 and WBP2. (B) COX-2 and IFITM3. (C) COX-2 and SFRP4. (D) WBP2 and IFITM3. (E) WBP2 and SFRP4. (F) SFRP4 and IFITM3.

(Fig. 2F). Proteoglycans in cancer adhesion, gene splicing, cancer-associated pathways, and cancer-associated miRNAs were the major enrichment pathways of the downregulated proteins (Fig. 2G). Either direct or indirect interactions among all differential proteins were plotted in the PPI network, and non-connected differential proteins were excluded (Figs. 2A–2C).

## IHC expression of WBP2, IFITM3, and SFRP4

WBP2 and SFRP4 were significantly expressed in the cytoplasm and cytosol of glandular cells in the ectopic endometrium of adenomyosis, but not in the myometrium and the mesenchymal cells of ectopic endometrium. IFITM3 was mainly expressed in mesenchymal cells of ectopic endometrium in adenomyosis, and to a lesser extent in glandular cells and the myometrium (Fig. 1A). Statistical analysis of the IHC results indicated the differences in the expression of WBP2 ($P = 0.016$), IFITM3 ($P = 0.025$), and SFRP4 ($P = 0.021$) were statistically significant (Fig. 1B).

## Correlation of COX-2, SFRP4, WBP2, and IFITM3 in ectopic endometrium of adenomyosis

In ectopic endometrium of adenomyosis, the expression levels of COX-2 were positively correlated with the expressions of WBP2, IFITM3, and SFRP4 ($r = 0.536$, $P < 0.01$; $r = 0.591$, $P < 0.01$; $r = 0.533$, $P < 0.01$; Figs. 3A–3C). Similarly, the expression of WBP2 was

**Table 3 The relationship between dysmenorrhea and expression of COX-2, WBP2, IFITM3, and SFRP4.**

| Group | Dysmenorrhea | | | $x^2$ | P |
|---|---|---|---|---|---|
| | No/mild $n$ (%) | Moderate $n$ (%) | Severe $n$ (%) | | |
| COX-2 | | | | | |
| LE | 7 (22.6) | 20 (64.5) | 4 (12.9) | 6.824 | 0.025 |
| HE | 0 (0.0) | 4 (44.4) | 5 (55.6) | | |
| WBP2 | | | | | |
| LE | 6 (23.1) | 16 (61.5) | 4 (15.4) | 2.808 | 0.300 |
| HE | 1 (7.1) | 8 (57.1) | 5 (35.7) | | |
| IFITM3 | | | | | |
| LE | 5 (41.7) | 6 (50.0) | 1 (8.3) | 6.597 | 0.033 |
| HE | 2 (7.1) | 18 (64.3) | 8 (28.6) | | |
| SFRP4 | | | | | |
| LE | 7 (26.9) | 16 (61.5) | 3 (11.5) | 7.510 | 0.020 |
| HE | 0 (0.0) | 8 (57.1) | 6 (42.9) | | |

Notes:
HE: high expression group
LE: low expression group.

**Table 4 The relationship between uterine volume and expression of COX-2, WBP2, IFITM3, and SFRP4.**

| Group | Uterine volume | | $x^2$ | P |
|---|---|---|---|---|
| | Normal $n$ (%) | Increscent $n$ (%) | | |
| COX-2 | | | | |
| LE | 22 (71.0) | 9 (29.0) | 0.216 | 0.642 |
| HE | 5 (55.6) | 4 (44.4) | | |
| WBP2 | | | | |
| LE | 19 (73.1) | 7 (26.9) | 0.452 | 0.501 |
| HE | 8 (57.1) | 6 (42.9) | | |
| IFITM3 | | | | |
| LE | 8 (66.7) | 4 (33.3) | 0.000 | 1.000 |
| HE | 19 (67.9) | 9 (32.1) | | |
| SFRP4 | | | | |
| LE | 19 (73.1) | 7 (26.9) | 0.452 | 0.501 |
| HE | 8 (57.1) | 6 (42.9) | | |

Notes:
HE: high expression group
LE: low expression group.

positively correlated with the expression of IFITM3 and SFRP4 (r = 0.461, $P < 0.01$; r = 0.576, $P < 0.01$; Figs. 3D and 3E). The expression levels of SFRP4 were positively correlated with that of IFTIM3 (r = 0.551, $P < 0.01$; Fig. 3F).

**Table 5 The relationship between menstrual blood volume and expression of COX-2, WBP2, IFITM3, and SFRP4.**

| Group | Menstrual blood volume | | $x^2$ | $P$ |
|---|---|---|---|---|
| | Normal $n$ (%) | Increscent $n$ (%) | | |
| COX-2 | | | | |
| LE | 9 (29.0) | 22 (71.0) | 0.00 | 1.000 |
| HE | 3 (33.3) | 6 (66.7) | | |
| WBP2 | | | | |
| LE | 6 (23.1) | 20 (76.9) | 0.884 | 0.347 |
| HE | 6 (42.9) | 8 (57.1) | | |
| IFITM3 | | | | |
| LE | 2 (16.7) | 10 (83.3) | 0.686 | 0.408 |
| HE | 10 (35.7) | 18 (64.3) | | |
| SFRP4 | | | | |
| LE | 8 (30.8) | 18 (69.2) | 0.000 | 1.000 |
| HE | 4 (28.6) | 10 (71.4) | | |

Notes:
HE: high expression group
LE: low expression group.

## The relationship between clinical characteristics and the expression of COX-2, WBP2, IFITM3, and SFRP4 in ectopic endometrium of adenomyosis

The relationship between the severity of dysmenorrhea and the expression of COX-2, IFITM3, SFRP4, and WBP2 were compared between the high expression group and the low expression group, and the differences were statistically significant for COX-2 ($P$ = 0.025), IFITM3 ($P$ = 0.033), and SFRP4 ($P$ = 0.020), but no significant difference was detected for WBP2 ($P$ = 0.300; Table 3). There was no statistically significant difference between the uterine volume of adenomyosis and the expression of COX-2 ($P$ = 0.642), WBP2 ($P$ = 0.501), IFITM3 ($P$ = 1.000), and SFRP4 ($P$ = 0.501) in the high expression group compared to the low expression group (Table 4). Similar results were observed between the distribution of menstrual volume and the expression of COX-2 ($P$ = 1.000), WBP2 ($P$ = 0.347), IFITM3 ($P$ = 0.408), and SFRP4 ($P$ = 1.000; Table 5).

## DISCUSSION

Adenomyosis is a fairly common gynecological disease, characterized mainly by the appearance of endometrial glands and mesenchymal cells in the myometrium. Although the pathogenesis of adenomyosis is unclear, the literature shows that invasion of the endometrial base membrane into the myometrium due to enhanced cell survival, EMT, and cell migration; continuous auto-microtrauma of the junctional zone; *de novo* metaplasia from adult stem cells and embryonic Mullerian remnants; and "from outside to inside invasion" are all associated with the development of adenomyosis (*Zhai et al., 2020*), with EMT playing a key role in adenomyosis progression (*Chen et al., 2020*). Exploring the

molecular mechanisms underlying the pathogenesis of adenomyosis may contribute to our understanding of its development.

The most important clinical manifestation of adenomyosis is dysmenorrhea, and inflammatory mediators are closely related to dysmenorrhea (*Carrarelli et al., 2017*; *Barcikowska et al., 2020*). Clinically, non-steroidal anti-inflammatory drugs (NSAIDs) can alleviate dysmenorrhea in adenomyosis by inhibiting COX-2 (*Marjoribanks et al., 2015*). In one study, inhibition of COX-2 expression significantly relieved dysmenorrhea symptoms, while overexpression of COX-2 was significantly correlated with dysmenorrhea symptoms in adenomyosis (*Li et al., 2019*). The results of the present study showed that the expression of COX-2 was associated with the severity of dysmenorrhea in adenomyosis. There are significant abnormalities in cell proliferation, apoptosis, and adhesion in adenomyosis tissues (*Zhai et al., 2020*; *Bourdon et al., 2021*). Inhibiting the expression of COX-2 can inhibit the survival, migration, and invasion of endometriotic cells, and promote cell apoptosis (*Banu et al., 2008*; *Knab, Grippo & Bentrem, 2014*). Therefore, it was hypothesized that COX-2 pathway-related differential proteins may be involved in these processes. In this study, GO and KEGG analyses showed that COX-2 pathway-related differential proteins might participate in processes such as substance localization and binding, cell adhesion, cell proliferation, and signal transduction. These proteins were mainly enriched in cancer-related pathways, apoptosis, and cellular energy metabolism.

COX-2 is a major inflammatory factor that induces cancer development. It is overexpressed in multiple cancer cells and is involved in the initiation of EMT (*Mattsson et al., 2015*; *Sicking et al., 2014*; *Dinicola et al., 2018*). In adenomyosis, COX-2 was highly expressed in ectopic endometrium compared to eutopic endometrium (*Li et al., 2019*; *Wang, Qu & Song, 2015*). Our previous research demonstrated that inhibiting COX-2 expression significantly reduced the infiltration of ectopic endometrium stromal in the myometrium of mice (*Liang et al., 2021*). In all molecular mechanisms of adenomyosis, EMT plays a pivotal part in the progression of adenomyosis (*Chen et al., 2020*). The Wnt/β-catenin signaling pathway is one of the more important pathways in EMT and has been demonstrated to be involved in the development of adenomyosis (*Yoo et al., 2020*). Higher expression levels of β-catenin were found in eutopic and ectopic endometrial tissue with adenomyosis compared to normal endometrium without adenomyosis. β-catenin is an important factor in the Wnt signaling pathway, downregulation of COX-2 can decrease the expression of β-catenin, and the expression of COX-2 is positively correlated with the expression of β-catenin (*Cai & Gao, 2021*). Therefore, COX-2 may participate in the EMT process in adenomyosis through the Wnt/β-catenin signaling pathway, and COX-2 may be a key influence in adenomyosis development.

WBP2 is an emerging tumor protein that functions in different oncogenic signaling pathways, such as the ER/PR, EGFR, PI3K, Hippo, and Wnt signaling pathways (*Tabatabaeian et al., 2020*), and WBP2 has gained attention as a potential drug target. Although the expression of WBP2 in adenomyosis tissue has not been reported, its expression and function in hepatocellular carcinoma and breast cancer have been confirmed (*Chen et al., 2017*; *Gao et al., 2020*). There is a higher expression of WBP2 in

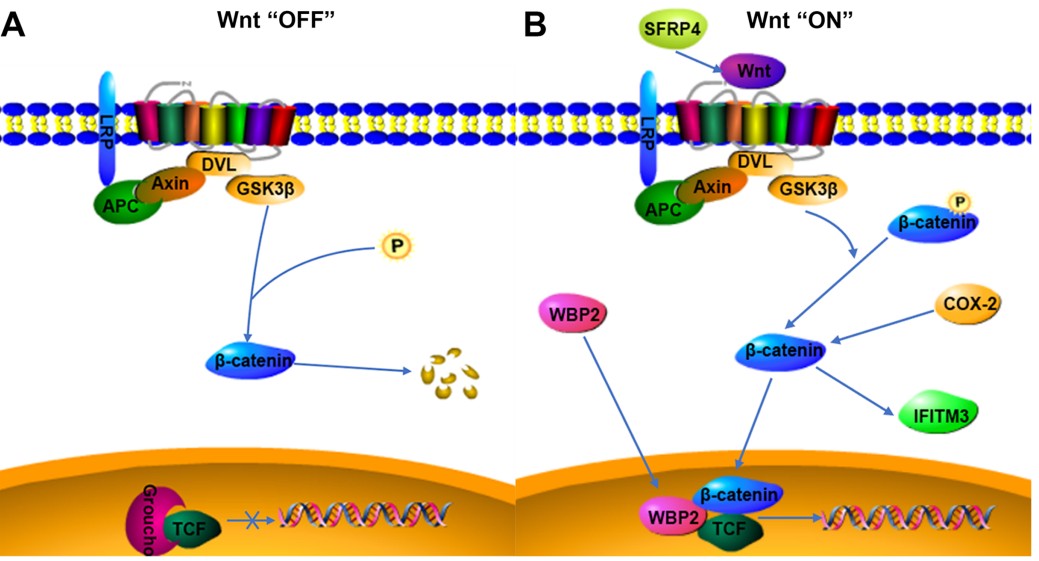

**Figure 4 Possible relationship of COX-2, WBP2, IFITM3, and SFRP4 with the Wnt/β-catenin pathway in adenomyosis.** (A) Wnt "OFF". In the absence of Wnt ligand, β-catenin is phosphorylated by the APC-Axin-GSK3β complex and degraded. The TCF binding to Groucho block the transcription of target genes. (B) Wnt "ON". When the Wnt ligand binds to its receptor, β-catenin is dephosphorylated and enters the nucleus. It then interacts with TCF to facilitate the transcription of target genes.

hepatocellular carcinoma cells than in normal liver cells, and downregulation of WBP2 blocks the activation of the Wnt/β-catenin signaling pathway (*Gao et al., 2020*). WBP2 is highly expressed in breast cancer tissues, and overexpression of WBP2 upregulates the ER and Wnt signaling pathways. When the Wnt/β-catenin pathway is activated, WBP2 can enter the nucleus with β-catenin and bind to TCF. These then act together to mediate target gene transcription (*Tabatabaeian et al., 2020*). In breast cancer tissues, the knockdown of WBP2 inhibits cell proliferation and promotes apoptosis (*Song et al., 2018*). WBP2 also regulates ER expression in an estrogen-dependent manner, and high ER expression and abnormal activation of the Wnt signaling pathway may be important factors in the pathogenesis of adenomyosis. In adenomyosis tissues, high expression of COX-2 can promote the upregulation of estrogen expression and also participate in EMT through the upregulation of β-catenin (*Hugo et al., 2015*; *Zhai et al., 2020*). WBP2 and COX-2 may be involved in the expression of the Wnt/β-catenin signaling pathway. The results of this study indicated that WBP2 expression was positively correlated with COX-2 in adenomyosis tissues. WBP2 expression was not significantly correlated with dysmenorrhea symptoms, while COX-2 was strongly associated with dysmenorrhea, therefore WBP2 may be regulated by the Wnt protein, but not by COX-2 (Fig. 4B).

IFITM3, a two-transmembrane-type protein that is closely associated with tumors, is strongly expressed in cancer. The proliferation, invasion, and migration of cancer cells can be significantly inhibited after silencing IFITM3 (*Zhao et al., 2013*). For example, in gastric cancer, knockdown of IFITM3 significantly inhibited the migration, invasion, and proliferation of tumor cells *in vitro* (*Hu et al., 2014*), and similar results were observed in

oral squamous cell carcinoma, colorectal cancer, breast cancer, glioma, and other tumors (*Li et al., 2011*; *Yang et al., 2013*; *Zhao et al., 2013*; *Gan et al., 2019*). The role of IFITM3 in adenomyosis is not yet clarified. However, normal endometrial tissue is negative for IFITM3, which is expressed in ovarian endometriosis lesions (*Fraunhoffer et al., 2015*), so IFITM3 may promote the development of endometriosis. In cancer tissues, IFITM3 promotes tumor migration and invasion, promotes EMT through the Wnt/β-catenin signaling pathway, and its overexpression leads to poor prognosis (*Hu et al., 2014*). β-catenin regulates the level of IFITM3, and downregulation of β-catenin leads to a decrease in IFITM3 expression (*Rajapaksa, Jin & Dong, 2020*), so IFITM3 is a downstream protein of β-catenin, and COX-2 is an upstream protein of β-catenin. β-catenin may be an intermediate protein between COX-2 and IFITM3, and IFITM3 may be regulated by the expression of COX-2 (Fig. 4B). The results of this study also showed that IFITM3 expression in ectopic endometrium of adenomyosis was positively correlated with COX-2. COX-2 is an important factor regulating the occurrence of dysmenorrhea symptoms, and IFITM3, as a downstream factor of COX-2, may be regulated by COX-2, so IFITM3 may be related to dysmenorrhea symptoms. The findings of this study also confirmed that IFITM3 expression in adenomyosis tissue was correlated with dysmenorrhea symptoms. IFITM3 may influence the development of adenomyosis and be a potential molecular target for the treatment of dysmenorrhea.

SFRP is an antagonist of Wnt ligands and is generally silent in cancer (*Bovolenta et al., 2008*; *Li et al., 2021*). SFRP4 is regulated by promoter methylation and its expression is downregulated when the SFRP4 gene is methylated (*Finch et al., 1997*; *Pawar & Rao, 2018*). Activation of the Wnt/β-catenin pathway is linked to tumorigenesis, and the SFRP4 protein can inhibit the Wnt pathway by binding to Wnt ligands and Frizzled receptors (*Ehrlund et al., 2013*; *Murakami et al., 2015*). In colon cancer and esophageal adenocarcinoma, SFRP4 gene methylation decreases the inhibition of the Wnt pathway (*Zou et al., 2005*). However, in gastric cancer, SFRP4 expression is upregulated by the reduced methylation, and high expression of SFRP4 could promote cell viability and proliferation and inhibit apoptosis, as well as activate the Wnt pathway and promote tumor progression (*Busuttil et al., 2021*). The role of SFRP4 in tumors is still highly controversial. SFRP4 is significantly expressed in the proliferative phase compared to the secretory phase in the endometrial cycle and is involved in endometrial physiology and carcinogenesis (*Abu-Jawdeh et al., 1999*). Aberrant activation of the Wnt/β-catenin signaling pathway is closely associated with the development of adenomyosis, so the role of SFRP4 in adenomyosis may be similar to that in gastric cancer, activating the Wnt signaling pathway and promoting cell proliferation, invasion, and metastasis (Fig. 4B). COX-2 expression was significantly correlated with dysmenorrhea and with invasive metastasis of ectopic endometrial cells. The findings of this study also indicated that SFRP4 was associated with dysmenorrhea symptoms and COX-2 was positively correlated with SFRP4, therefore the two factors may interact and participate in the development of adenomyosis through the Wnt/β-catenin signaling pathway, though the relationship between them needs further study.

The development of adenomyosis is a complex process involving possible mechanisms such as angiogenesis and neurofibrillogenesis, chronic inflammatory stimulation, and abnormal cell proliferation (*Vannuccini et al., 2017*). COX-2, WBP2, IFITM3, and SFRP4 are associated with cell proliferation and inflammation in a variety of cancers (*AlAshqar et al., 2021*; *Tabatabaeian et al., 2020*; *Lee, 2022*; *Busuttil et al., 2021*). Their expression may promote the development of adenomyosis, but may only be involved in the early stages of the disease, whereas increased menstrual bleeding and increased uterine volume in patients with adenomyosis are late events that occur after the development of adenomyosis. Thus, the expression of COX-2, WBP2, IFITM3, and SFRP4 did not correlate significantly with menstrual flow and volume in adenomyosis. If targeted therapy of related factors is used in the early stages of adenomyosis, it might be possible to inhibit the occurrence of late uterine enlargement and heavy menstrual bleeding.

## CONCLUSIONS

This study demonstrated that COX-2, WBP2, IFITM3, and SRFP4 may be involved in the pathogenesis of adenomyosis. COX-2, IFITM3, and SFRP4 are significantly associated with the degree of dysmenorrhea, and they may be potential molecular targets for the treatment of dysmenorrhea in patients with adenomyosis. Further experiments are needed to verify this possibility. This study provides a theoretical basis for targeted treatment of dysmenorrheal in patients with adenomyosis. However, this study had some limitations. The number of adenomyosis cases was small and the data need to be verified in a larger sample size. Further studies are also required to validate the underlying molecular mechanism between COX-2, IFITM3, SFRP4 and adenomyosis.

## ACKNOWLEDGMENTS

We are grateful to all those who have contributed to this research.

### Funding

This work was supported by the Henan Science and Technology Project (23A320001), the Henan Health Commission Medical Science and Technology Project (2018020167). The funders had no role in study design, data collection and analysis, decision to publish, or preparation of the manuscript.

### Grant Disclosures

The following grant information was disclosed by the authors:
Henan Science and Technology Project: 23A320001.
Henan Health Commission Medical Science and Technology Project: 2018020167.

### Competing Interests

The authors declare that they have no competing interests.

## Author Contributions

- Jihua Zhang performed the experiments, analyzed the data, prepared figures and/or tables, authored or reviewed drafts of the article, and approved the final draft.
- Luying Shi performed the experiments, analyzed the data, authored or reviewed drafts of the article, and approved the final draft.
- Jingya Duan performed the experiments, analyzed the data, authored or reviewed drafts of the article, and approved the final draft.
- Minmin Li performed the experiments, analyzed the data, authored or reviewed drafts of the article, and approved the final draft.
- Canyu Li conceived and designed the experiments, authored or reviewed drafts of the article, and approved the final draft.

## Human Ethics

The following information was supplied relating to ethical approvals (*i.e.*, approving body and any reference numbers):

The Third Affiliated Hospital Zhengzhou University granted Ethical approval to carry out the study in its hospitals.

## Data Availability

The bioinformatics analysis data is available at Zenodo: Jihua Zhang, Luying Shi, Jingya Duan, Minmin Li, & Canyu Li. (2023). Proteomic detection of COX-2 pathway-related factors in patients with adenomyosis [Data set]. Zenodo. https://doi.org/10.5281/zenodo.8281116.

The patient tissue data are in the Supplemental File.

## Supplemental Information

Supplemental information for this article can be found online at http://dx.doi.org/10.7717/peerj.16784#supplemental-information.

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
