# Peer review of "Proteomic detection of COX-2 pathway-related factors in patients with adenomyosis"

_PeerJ, doi:10.7717/peerj.16784_

## Round 0.1 · original submission · Minor Revisions

Please address all the comments and looking forward to a revised version. I would like to have a thorough English revision before submission. I would consider the suggestions provided by reviewers as minor and appreciate your efforts to resubmit at the earliest.

**Language Note:** The Academic Editor has identified that the English language must be improved. PeerJ can provide language editing services - please contact us at copyediting@peerj.com for pricing (be sure to provide your manuscript number and title). Alternatively, you should make your own arrangements to improve the language quality and provide details in your response letter. – PeerJ Staff

·

Basic reporting

Context of this paper is well-written and easy to approach. However, the following contents will be benefit from revision:
a. Discussion is poorly organized. Current version verbally discussed the relationship between IFITM3, SFRP4, WBP2 and COX-2. It is hard for reader to follow and a diagram showing the relationship will be clear to the readers.
b. Tables miss content behind “2”

Experimental design

The proteomic and IHC experiments have high quality. However, simply points out the correlation between COX-2 and IFITM3, SFRP4, WBP2 is not enough to suggest the use of them as biomarkers. The conclusion is too strong. If author really wants to make this conclusion, a siRNA experiments could be considered to knockdown these three to show effect. Otherwise, authors have to soften their tone to make such conclusion.

Validity of the findings

As mentioned in the above section, overall data has good quality but authors over-state the conclusion from this study. Additional experiments are needed

Additional comments

Overall this is an interesting study with good quality of data. However, this paper can be better organized by introducing diagram showing relationship between COX-2 and IFITM3, SFRP4, WBP2. Discussion part could be better organized and increase more content to show how this proposed study can benefit the clinical practice. Additional experiments are needed to make current conclusions (siRNA, etc.)¬

·

Basic reporting

1. The quality of English writing is acceptable as an initial submission but needs to be significantly improved to meet publication standards. There are a number of grammatical errors. Below are a few examples where the language could be improved. Language editing is required throughout the manuscript, extending beyond these examples.

Lines 50-52 should be “Our previous studies have also found that COX-2 was overexpressed in the ectopic gland and stroma rather than the eutopic endometrium. Silencing COX-2 expression can inhibit stromal cell proliferation and invasion.”

Lines 56-57 should be “However, the mechanism by which COX-2 affected endometrial cells and the factors with which it acted are unclear.”

Line 63: “So that…” can be revised to “This study aimed to discover target genes associated with adenomyosis, serving as indicators for early prediction or condition monitoring, as well as potential targets for treatment.”

Line 153: “Identificational" is not a standard English word.

Line 159: “Highly” should be “high”.

2. The PPI network illustration in Fig. 1E contains too many components, making it impossible to read. The authors may want to consider presenting multiple networks, each with fewer proteins, to enhance readability.

Experimental design

1. The primary design of the study involves comparing proteins from COX-2 high and low patient samples. However, the rationale supporting this experimental design is inadequately provided. While some literature related to COX-2 was referenced, the clinical significance of varying COX-2 levels in different patients was not introduced. If this information has not been reported previously, the authors should propose their hypothesis that led to this experimental design.

2. Line 151: What are the criteria for selecting five samples from each group? If it was randomly selected, please make it clear.

Validity of the findings

For each figure or table that includes statistical analysis, please elaborate on the statistical method, including details on sample size and the chosen test. The present manuscript lacks the necessary information to be considered statistically sound.

Additional comments

This manuscript submitted by Zhang et al. used quantitative proteomics to study adenomyosis patient specimens. By comparing proteomic data between COX-2 high and low samples, the authors identified a number of differentially expressed proteins that may relate to the COX-2 pathway and may serve as potential biomarkers. Before acceptance for publication, the comments are expected to be adequately addressed.

---

## Round 0.2 · accepted · Accept

The following paper is accepted upon reviewer recommendations.

·

Basic reporting

My part of concern has been correctely addressed. Thanks for the effort.

Experimental design

My part of concern has been correctely addressed. Thanks for the effort.

Validity of the findings

My part of concern has been correctely addressed. Thanks for the effort.

Additional comments

My part of concern has been correctely addressed. Thanks for the effort.

·

Basic reporting

The writing and figure organization have been significantly improved.

Experimental design

The authors have addressed previous comments on experimental design.

Validity of the findings

The authors have addressed previous comments on statistical methods.